# The Role of Vascular Cell Adhesion Molecule-1 (VCAM-1) in Predicting Complicated Appendicitis in Children

**DOI:** 10.3390/diagnostics14121256

**Published:** 2024-06-14

**Authors:** Wen-Ya Lin, En-Pei Lee, Chun-Yu Chen, Bei-Cyuan Guo, Mao-Jen Lin, Han-Ping Wu

**Affiliations:** 1Department of Pediatrics, Taichung Veterans General Hospital, Taichung 40705, Taiwan; wylin002@gmail.com; 2Division of Pediatric Critical Care Medicine, Department of Pediatrics, Chang Gung Memorial Hospital at Linko, Kweishan, Taoyuan 33305, Taiwan; pilichrislnp@gmail.com; 3College of Medicine, Chang Gung University, Taoyuan 33302, Taiwan; 4Department of Emergency Medicine, Tungs’ Taichung MetroHarbor Hospital, Taichung 43503, Taiwan; yoyo116984@gmail.com; 5Department of Nursing, Jen-Teh Junior College of Medicine, Nursing and Management, Miaoli 356006, Taiwan; 6Department of Pediatrics, National Cheng Kung University Hospital, College of Medicine, National Cheng Kung University, Tainan 70403, Taiwan; gbc628@gmail.com; 7Division of Cardiology, Department of Medicine, Taichung Tzu Chi Hospital, The Buddhist Tzu Chi Medical Foundation, No. 88, Sec. 1, Fong-Sing Rd., Tanzi District, Taichung 427213, Taiwan; 8Department of Medicine, College of Medicine, Tzu Chi University, Hualien 970374, Taiwan; 9Department of Pediatrics, Chiayi Chang Gung Memorial Hospital, No. 6, W. Sec., Jiapu Rd., Puzi City 61363, Taiwan

**Keywords:** vascular cell adhesion molecule-1, VCAM-1, appendicitis, complicated appendicitis, children

## Abstract

Background: Acute appendicitis is a common abdominal emergency observed in emergency departments (ED). Distinguishing between uncomplicated and complicated appendicitis is important in determining a treatment strategy. Serum soluble vascular cell adhesion molecule-1 (VCAM-1) is an inflammatory biomarker. We aimed to determine the role of VCAM-1 in predicting complicated appendicitis in children. Methods: Pediatric patients with suspected appendicitis admitted to the ED were enrolled in this prospective study. Pre-surgical serum VCAM-1 was tested in children with acute appendicitis within 72 h of symptoms (from day 1 to day 3). Serum VCAM-1 levels were further analyzed and compared between patients with and without complicated appendicitis. Results: Among the 226 pediatric appendicitis patients, 70 had uncomplicated appendicitis, 138 had complicated appendicitis, and 18 had normal appendices. The mean serum VCAM-1 levels in patients with perforated appendicitis were higher than in those with simple appendicitis (*p* < 0.001). On day 1 to day 3, the mean VCAM-1 levels in patients with complicated appendicitis were all significantly higher than in those with uncomplicated appendicitis (all *p* < 0.001). Conclusion: Serum VCAM-1 levels may be helpful in differentiating uncomplicated and complicated appendicitis in children and could predict appendiceal perforation.

## 1. Introduction

Acute appendicitis (AA) is a common cause of acute abdominal pain in emergency care, and lifetime risks vary geographically, ranging from 16% in South Korea to 9% in the USA and 2% in Africa [1]. The most common reason for abdominal surgery in children is appendicitis. Typical clinical presentation includes the presence of initial peri-umbilical abdominal pain with migration to the right iliac fossa, anorexia, fever, and localized guarding and tenderness of the right iliac fossa area. However, younger children often have atypical presentations [2]. Symptoms may also be vague with a poor description in children, rendering diagnosis with history and physical examination more challenging [3]. Male predominance has also been reported with a male-to-female ratio of 1.4:1 [4].

Acute appendicitis may be classified as uncomplicated appendicitis and complicated appendicitis. Simple or uncomplicated appendicitis is defined as an inflamed appendix without necrosis or perforation [5]. Spontaneous resolution of uncomplicated appendicitis is not uncommon, and perforation can rarely be prevented as most perforation occurs before arrival at a hospital [6]. Recent studies have revealed non-surgical antibiotic treatment for patients with uncomplicated appendicitis to be effective; Ref. [7] spontaneous resolution without antibiotic use has also been reported [8]. In contrast, patients with complicated appendicitis require emergency surgical treatment, with the exception of patients with a periappendicular abscess [9]. Thus, there has been a shift to early differentiation and accurate pre-operative diagnosis of uncomplicated and complicated appendicitis in clinical practice.

Traditionally, AA is thought to be initiated by an intra-luminal obstruction, resulting in impaired blood and lymphatic flow with localized edema. Thereafter, appendiceal epithelium barrier function is impaired with bacterial invasion into the submucosa. The immune system is activated with the release of cytokines and chemokines, with local recruitment of T cells, monocytes, and natural killer cells. Further decreased blood flow leads to hypoxia, with ensuing necrosis and eventual appendiceal perforation [10]. The notion that an initial intra-luminal obstruction (with fecalith being the most implicated culprit) serves as a trigger for subsequent inflammation in AA pathogenesis has been challenged. The main reason that this concept has been abandoned is that fecaliths are only found in a minority of patients with confirmed AA. In fact, the presence of a fecalith can also be found in patients without symptoms and signs of AA [11]. The etiology of AA is, thus, more complex and may be multifactorial.

The degree of severity of inflammation ultimately differentiates and determines the clinical presentation of uncomplicated and complicated appendicitis. This is due to the complex interplay of inflammatory cells, cytokines, and chemokines. Milder inflammation with uncomplicated appendicitis may require the use of antibiotics or it may resolve spontaneously. The increased severity of appendicitis inflammation further aggravates edema and ischemia, leading to blood vessel thrombosis, and the weakening and necrosis of epithelial walls, which could result in perforation of the appendix [1,3,11].

Classically, the rapid diagnosis of acute appendicitis is commonly based on a brief detailed history, a focused physical examination, and directed laboratory findings that consist of clinical diagnostic scoring systems [12]. The Alvarado score, appendicitis inflammatory response score, and the pediatric appendicitis score have been clinically applied for diagnosing appendicitis and may be helpful for clinical application [13,14,15]. Recently, laboratory markers in differentiation between uncomplicated and complicated appendicitis have been explored. Specific to pediatric appendicitis, systemic reviews and meta-analyses performed have suggested the platelet-to-lymphocyte ratio [16], higher serum total bilirubin level [17], and elevated serum interleukin-6 [18] may be of value in early discrimination of complicated appendicitis. Significantly lower levels of sodium were also found in children with complicated appendicitis [19].

Vascular cell adhesion molecule-1 (VCAM-1) is a major regulator of leukocyte adhesion and trans-endothelial migration. This is facilitated by the endothelial surface VCAM-1 interaction with α4β1-integrin expressed on leukocytes, which activates the signaling pathway, allowing the final trans-endothelial migration of leukocytes [20]. VCAM-1 thus plays an integral role during this acute inflammatory process and may influence the final clinical presentation of uncomplicated or complicated appendicitis.

Traditionally, emergent appendectomy is advocated for all patients with AA due to concerns about impending rupture. It was previously thought that inevitable uncomplicated appendicitis would inevitably progress to perforation, but recent evidence indicates AA largely exists as two separate clinical entities, i.e., uncomplicated and complicated appendicitis, with different recommended treatment regimens [21,22]. Emphasis must, therefore, be placed on the correct diagnosis rather than rapid diagnosis. Guidelines are provided regarding the treatment of AA. Emergent surgical treatment for patients (adults and children) with complicated appendicitis is recommended, whereas antibiotics alone can be given for uncomplicated appendicitis [9,23]. In children with complicated appendicitis, early appendectomy is the best management, as children with late appendectomy were shown to be more likely to have at least one complication in comparison to children with early appendectomy (*p* < 0.01) [24]. Antibiotics as the initial treatment for children with uncomplicated appendicitis may be feasible and effective without increased complication risks [25,26]. However, the same guidelines advising of the methods for pre-operative differentiation between uncomplicated and complicated appendicitis are not as well defined.

Thus, this study aimed to explore the role of serum VCAM-1 in differentiating pre-operative uncomplicated and complicated appendicitis in children presenting to the emergency department (ED), and the changes in VCAM-1 level from duration of symptom onset (day 1 to day 3) in children with AA are presented.

## 2. Materials and Methods

### 2.1. Patient Population

Pediatric patients with suspected acute appendicitis aged 4–18 years were included in this prospective study from August 2016 to July 2019. The enrollment criteria included symptoms and signs that are characteristic of appendicitis, such as abdominal pain, anorexia, nausea, vomiting, fever, migratory abdominal pain, and right lower quadrant rebound tenderness. Children with symptoms and signs lasting > 3 days were excluded from the study. Subsequent diagnosis of appendicitis was confirmed by histopathological examination of the surgical specimen. Complicated appendicitis was defined as gangrenous or perforated appendicitis.

The study protocol was approved by the Institutional Review Board of Chang Gung Memorial Hospital (IRB No.: 1049627A3, approval date: 1 August 2016) and the ethics committee, and was conducted in accordance with the Declaration of Helsinki. Informed consent was obtained from the parents and/or legal guardians of the children who participated in the study. All methods were performed in accordance with the relevant protocols.

### 2.2. Study Design

On admission, basic parameters, including age, sex, body temperature, time of onset of symptoms and signs, time of admission, and management, were recorded. Blood samples were obtained from all patients for further analysis. Serum vascular cell adhesion molecule-1 (VCAM-1) was tested within 72 h of symptom onset; VCAM-1 measurement was simultaneously measured with the chemokine biochip array using Evidence Investigator (Randox Laboratories Ltd., Crumlin, County Antrim, United Kingdom), a semiautomated software program. The duration was designated as <24 h on day 1, 24–48 h on day 2, and 48–72 h on day 3. Further histological subdivision of the excised surgical specimens into simple appendicitis (SA), gangrenous appendicitis (GA), and perforated appendicitis (PA) was performed. SA was defined as neutrophil infiltration of the mucosa, submucosa, or muscularis propria. Transmural inflammation with necrosis and mucosal ulceration were observed in GA. PA was defined as extensive transmural inflammation with perforation.

### 2.3. Treatment Outcome

All patients received appendectomies for appendicitis. Serum VCAM-1 levels were compared among patients with SA, GA, and PA, as well as between uncomplicated and complicated appendicitis, based on the different times of presentation of appendicitis.

### 2.4. Statistics

Statistical analysis of VCAM-1 was carried out in each group using the Mann–Whitney U test, chi-square test, one-way analysis of variance (ANOVA), logistic regression analysis, and the receiver operating characteristic (ROC) curves. Values are presented as a median (25th and 75th percentiles). In the ROC curves, the test characteristics of these different cutoff values, including sensitivity, specificity, positive likelihood ratio (LR+), negative likelihood ratio (LR−), and area under the ROC curve (AUC), were also examined. Statistical significance was defined as *p* < 0.05 and all statistical analyses were conducted using IBM SPSS Statistics software (version 22.0; SPSS Inc., Chicago, IL, USA).

## 3. Results

During this study period, a total of 226 children who underwent surgical intervention had histologically confirmed appendicitis (Figure 1). These included 70 with uncomplicated appendicitis and 138 with complicated appendicitis, and 18 had normal appendicitis. Among the 138 children with complicated appendicitis, 36 children had GA, and 102 children had PA. The study group comprised 135 males (59.7%) and 91 females (40.3%), with a mean age of 10.0 ± 4.5 years. The patients’ characteristics in two groups (non-PA and PA) are shown in Table 1. In clinical symptoms and signs, migration pain, vomiting, fever, right lower quadrant (RLQ) pain, and rebound pain were significant parameters between children with PA and those with non-perforated appendicitis. Fever, RLQ pain, and rebounded pain were more common in patients with PA while migration pain and vomiting were more common in patients with non-perforated appendicitis. In addition, in routine laboratory tests, only CRP was a significant parameter between children with PA and those with non-perforated appendicitis. Neither white blood cells or neutrophils showed any significant differences between the two groups.

In the SA group, the serum VCAM-1 levels remained markedly lower than those in the GA and PA groups. The PA group had the highest VCAM-1 level among the 3 groups. The serum VCAM-1 levels in children with GA and PA were significantly elevated, with mean levels more than 17 times higher than the level found in children with SA. Moreover, a consistent trend of VCAM-1 levels between uncomplicated and complicated appendicitis with further subdivisions according to the clinical course (days 1–3) is listed in Table 2. From day 1 to day 3, the median VCAM-1 levels in children with complicated appendicitis were all significantly higher than in those with uncomplicated appendicitis (all *p* < 0.001). Table 3 shows the cutoff values of VCAM-1 to confirm or exclude the status of complicated appendicitis on day 1 to 3. The cutoff values for VCAM-1 of 1500 pg/mL on day 1, 1000 pg/mL on day 2, and 300 pg/mL on day 3 had the highest sensitivity (1.00), and the highest specificity (1.00), to predict complicated appendicitis.

## 4. Discussion

In this study, serum VCAM-1 was significantly and markedly elevated in complicated appendicitis from day 1 to day 3 of clinical disease compared to those with uncomplicated appendicitis. Moreover, serum VCAM-1 was highly elevated in children with appendiceal perforation, followed by GA and SA.

Increased mortality and morbidity rates are also associated with perforated appendicitis in children. Mortality risk is less than 0.1% in simple appendicitis, 0.6% in gangrenous appendicitis, and 5% in perforated appendicitis [9,27]. Recently, interest has been raised on whether non-surgical treatment for uncomplicated appendicitis in adult and pediatric patients is a safe and effective option, while complicated appendicitis is managed with emergent surgery [25,28,29,30]. Therefore, it is necessary to identify cases of uncomplicated and complicated appendicitis. In this study, serum VCAM-1 was consistently and significantly elevated in complicated appendicitis from day 1 to day 3 of clinical disease compared to those with uncomplicated appendicitis. Moreover, the serum VCAM-1 levels were highly elevated in children with appendiceal perforation, followed by those with GA and SA. SA can progress to RA. As appendicitis progresses, local inflammation of the appendiceal area can also progress to systemic inflammation.

The inflammatory response in acute appendicitis is regulated by a multistep cascade that includes immune cell recruitment, release of various mediators (selectins and cell adhesion molecules), and interactions between leukocytes and the endothelium [31]. VCAM-1 plays an important role in this acute inflammatory response, and recent evidence has revealed another role in dysregulated para-inflammation, which is closely associated with the progression of various immune diseases (rheumatoid arthritis, asthma, and transplant rejection) and cancer [32,33]. Two different immune responses and inflammatory patterns corresponding to uncomplicated and complicated appendicitis have been described. Those with complicated appendicitis have inflammatory pathways mediated predominantly by innate immunity, shorter history, and higher inflammatory scores [34]. Acute inflammatory response in acute appendicitis is regulated by a multi-step cascade, including immune cell recruitment, release of various mediators (selectins and cell adhesion molecules), and interactions between leukocytes and the endothelium [31]. VCAM-1 plays an important role in this acute inflammatory response and elevated levels may correlate with a higher degree of inflammation and complicated appendicitis.

In this study, elevated serum levels of VCAM-1 were demonstrated to be a strong predictor of complicated appendicitis within the first 3 days of the disease course. Currently, few predictors for complicated appendicitis require the use of either multiple parameters or imaging modalities, which have their limitations and disadvantages. Ultrasound studies are operator dependent, CT requires radiation dose exposure, and MRI is of limited use due to its cost and availability. These imaging modalities also require some levels of patient cooperation, and sedation may be required in certain selected pediatric cases, which may increase sedative drug exposure, increase patient discomfort, induce parent/caregiver anxiety, and delay diagnosis time. Biomarkers in the blood can be drawn in the ED when an intravenous line is applied and does not add further discomfort to the patient/family. Biomarkers (routine and novel) in combination with other clinical variables could be used to assist in the pre-operative diagnosis of complicated appendicitis in children. Although novel biomarkers (IL-6, neutrophil gelatinase-associated lipocalin [35], pentraxin-3 [36], leucine-Rich α-2-glycoprotein 1 [37], hyponatremia [19], hyperbilirubinemia [38], hyperfibrinogenemia [39], and ischemia-modified albumin [40]) in the identification of complicated appendicitis has shown promising results, they are not yet widely accessible in clinical practice and much work is still warranted to validate clinical accuracy and improve accessibility. Cost is also a major deterrence to wide clinical application. However, these should not discourage further studies, as pre-operative diagnosis of complicated appendicitis is important in clinical decision making and ensuring the delivery of accurate high-quality care to children.

There were several limitations in this study. The sample size was small, all patients were from a single hospital, and patients’ serum VCAM-1 was sampled at only one time point over a 3-day period. VCAM-1 is not yet widely accessible in clinical practice and its use in practical terms may be limited by cost and availability.

## 5. Conclusions

VCAM-1 could predict appendiceal perforation and may serve as a useful predictor in differentiating complicated from uncomplicated appendicitis in children.

## Figures and Tables

**Figure 1 diagnostics-14-01256-f001:**
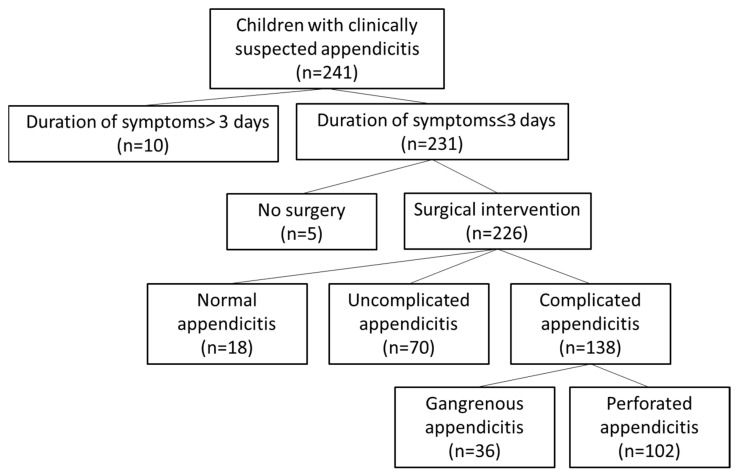
Workflow of patient population.

**Table 1 diagnostics-14-01256-t001:** Demographics between children with a non-perforated appendix and perforated appendix.

Variable	Non-Perforated Appendix (N = 106)	Perforated Appendix (N = 102)	*p* Value
Age (year)	9.8 ± 3.7	10.1 ± 3.9	0.570
Gender (male:female)	64:42	51:51	0.132
Clinical symptoms			
Migration pain	65 (61.3%)	42 (41.2%)	0.004
Vomiting	61 (57.1%)	50 (49.0%)	0.218
Fever	49 (46.2%)	58 (56.9%)	0.125
RLQ pain	104 (98.1%)	102 (100.0%)	0.163
Rebound pain	88 (83.0%)	96 (94.1%)	0.012
Laboratory examination			

WBC, 10^3^ uL	16.8 ± 5.8	16.4 ± 6.4	0.637
Neutrophils, %	82.0 ± 10.5	80.9 ± 8.7	0.413
CRP, mg/L	125 ± 95.6	188.3 ± 105.8	<0.001

**Table 2 diagnostics-14-01256-t002:** Comparison of VCAM-1 levels in children with uncomplicated and complicated appendicitis from days 1 to 3.

Duration	VCAM-1 (pg/mL)		*p* Value
	Uncomplicated Appendicitis	Complicated Appendicitis	
	Median (25–75%) (No.)	Median (25–75%) (No.)	
Day 1	61(60–68) (30)	3506 (3368–3619) (12)	<0.001
Day 2	59(59–61) (28)	3010 (2666–3308) (84)	<0.001
Day 3	60(60–62) (12)	3134 (2227–3491) (42)	<0.001

VCAM-1: vascular cell adhesion molecule-1.

**Table 3 diagnostics-14-01256-t003:** Sensitivity, specificity, LR+, and LR− at the best cutoff values of VCAM-1 to rule in complicated appendicitis in children from days 1 to 3.

VCAM-1		Cutoff Value (pg/mL)	Sensitivity	Specificity	LR+	LR−	AUC
	Day 1	1500	1.000	1.000	-	0.000	1.000
	Day 2	1000	1.000	1.000	-	0.000	1.000
	Day 3	300	1.000	1.000	-	0.000	1.000

VCAM-1: vascular cell adhesion molecule-1.

## Data Availability

The original contributions presented in the study are included in the article, further inquiries can be directed to the corresponding authors.

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
