# Peer review of "The Role of Vascular Cell Adhesion Molecule-1 (VCAM-1) in Predicting Complicated Appendicitis in Children"

_diagnostics, 2024, doi:10.3390/diagnostics14121256_

Round 1

Reviewer 1 Report

Comments and Suggestions for Authors

The authors investigated the role of Vascular Cell Adhesion Molecule-1 (VCAM-1) in predicting complicated appendicitis in children and concluded that VCAM-1 levels could be helpful in differentiating between uncomplicated and complicated appendicitis in children and could predict appendiceal perforation.

Significant objections were raised regarding the study design and methodology and the presentation of the results. My comments are as follows:

1. Abstract – The results section in the abstract should be better presented. The authors should provide the exact values of the variables studied. Also, exact p-values should be given for each variable (instead of p<0.05). The same applies to the measurements on days 1 to 3.

2. Introduction – An introduction is very extensive and should be shortened and focused on the main objectives of the study. General statements and repetition of known facts about acute appendicitis should be omitted. Some rare causes such as blunt abdominal trauma or seasonal variations of acute appendicitis should be removed, as well.

3. Although the introduction is very extensive, the authors failed to present the most important – the role of inflammatory markers in acute appendicitis. The authors should add more data regarding biomarkers for acute appendicitis. Recently, several biomarkers have been used to differentiate between perforated and non-perforated acute appendicitis. Authors should add several sentences and appropriate references. In addition to the standard biomarkers, there are several new biomarkers for acute appendicitis, such as hyperfibrinogenemia (doi: 10.1111/ans.13316), ischemia modified albumin (doi: 10.1016/j.ajem.2016.10.010), pentraxin-3 (PTX-3) (doi: 10.1016/j.ajem.2019.01.010), hyperbilirubinemia (doi: 10.1089/sur.2021.107), neutrophil gelatinase-associated lipocalin (NGAL) or interleukin-6 (IL-6) (doi: 10.1007/s00383-020-04650-y), hyponatremia (10.3390/children9071070) and leucine-Rich α-2-glycoprotein 1 (LGR-1) (doi: 10.3390/jcm12072455), have recently been investigated. These biomarkers showed good predictive values for the detection of acute appendicitis and the differentiation between complicated and simple acute appendicitis.

4. In their introduction, the authors state that luminal obstruction can be caused by fecal matter, lymphoid hyperplasia, fibrous bands, cecal carcinoma, or other mass lesions. I would recommend rephrasing this sentence as the majority of cases are not caused by cecal carcinoma but by neuroendocrine tumors of the appendix. In addition, foreign bodies can also be the cause of acute appendicitis. Please revise the article and add appropriate references.

5. Methodology – The authors state that the prospective study was conducted from 2016 to 2019. Please provide the exact time period (month and year).

6. In addition to the previous comment that almost five years have passed since the results were finalized, why were the results not published earlier?

7. The authors state that the study protocol was approved by the Institutional Review Board of Chang Gung Memorial Hospital (IRB No.: 1049627A3). Please add the date of approval next to the approval number.

8. Please add a new paragraph in methodology (Outcomes of treatment) and describe the primary and secondary outcomes of the study.

9. It is unclear which statistical test was used to test the normality of the data distribution. Please clarify.

10. The main methodological objection to this study is the lack of sample size calculation. As this is a prospective study, sample size calculation is mandatory. Please comment and clarify this point! If this was not done, the authors should clearly state why and add this to the study limitations. If a sample size calculation was performed, please include this as a separate paragraph in the methodology.

11. The next important objection is the presentation of the baseline characteristics of the patients. The baseline demographics of the patients and a comparison of the patients in terms of baseline demographics and other inflammatory biomarkers should be presented. It is completely inappropriate to present only the results of the biomarkers studied without providing the baseline data of the patients. Demographic data such as age, gender, comorbidities, weight, height, BMI, clinical data (pain, vomiting, body temperature, RLQ pain, rebound sensitivity) and laboratory inflammatory markers (WBC, CRP and neutrophils) should be compared between groups. In addition, these variables should also be explained in the methodology.

12. Tables 1 and 2 – Exact p-values (instead of p<0.05) should be reported.

13. The discussion is extremely extensive with many repetitions of unnecessary general data. The discussion need to be totally reorganized / revised. There is a lack of coherent comparison of the study results with the existing literature, focusing on the main objectives (especially in regards to investigated marker). Do not provide an overview of the literature in this section. Do not discuss your findings piecemeal. Focus on the results of the main objectives of the study. Write in four consecutive paragraphs (without headings) (a) summary (no data) of the findings of this study; (b) logical and coherent comparison with the existing literature focusing on the main aims; (c) limitations of the study; and (d) implications for practice/policy/research with a concluding statement.

14. The quality of the English language should be improved. The study would benefit from professional editing.

Comments on the Quality of English Language

The quality of the English language should be improved. The study would benefit from professional editing.

Author Response

Reviewer 1

Comments and Suggestions for Authors

The authors investigated the role of Vascular Cell Adhesion Molecule-1 (VCAM-1) in predicting complicated appendicitis in children and concluded that VCAM-1 levels could be helpful in differentiating between uncomplicated and complicated appendicitis in children and could predict appendiceal perforation.

Significant objections were raised regarding the study design and methodology and the presentation of the results. My comments are as follows:

  1. Abstract – The results section in the abstract should be better presented. The authors should provide the exact values of the variables studie Also, exact p-values should be given for each variable (instead of p<0.05). The same applies to the measurements on days 1 to 3.

Answer: Thank you for your suggestions. The p-values were very low, so we just presented p<0.001.

  1. Introduction – An introduction is very extensive and should be shortened and focused on the main objectives of the study. General statements and repetition of known facts about acute appendicitis should be omitted. Some rare causes such as blunt abdominal trauma or seasonal variations of acute appendicitis should be removed, as well.

Answer: Thank you for your suggestions. We have removed statements in our revised manuscript.

  1. Although the introduction is very extensive, the authors failed to present the most important – the role of inflammatory markers in acute appendicitis. The authors should add more data regarding biomarkers for acute appendicitis. Recently, several biomarkers have been used to differentiate between perforated and non-perforated acute appendicitis. Authors should add several sentences and appropriate references. In addition to the standard biomarkers, there are several new biomarkers for acute appendicitis, such as hyperfibrinogenemia (doi: 10.1111/ans.13316), ischemia modified albumin (doi: 1016/j.ajem.2016.10.010), pentraxin-3 (PTX-3) (doi: 10.1016/j.ajem.2019.01.010), hyperbilirubinemia (doi: 10.1089/sur.2021.107), neutrophil gelatinase-associated lipocalin (NGAL) or interleukin-6 (IL-6) (doi: 10.1007/s00383-020-04650-y), hyponatremia (10.3390/children9071070) and leucine-Rich α-2-glycoprotein 1 (LGR-1) (doi: 10.3390/jcm12072455), have recently been investigated. These biomarkers showed good predictive values for the detection of acute appendicitis and the differentiation between complicated and simple acute appendicitis.

Answer: Thank you for your suggestions. We have added statements in our revised manuscript.

  1. In their introduction, the authors state that luminal obstruction can be caused by fecal matter, lymphoid hyperplasia, fibrous bands, cecal carcinoma, or other mass lesions. I would recommend rephrasing this sentence as the majority of cases are not caused by cecal carcinoma but by neuroendocrine tumors of the appendix. In addition, foreign bodies can also be the cause of acute appendicitis. Please revise the article and add appropriate references. Answer: Thank you for your suggestions. We have removed statements in our revised manuscript.
  2. Methodology – The authors state that the prospective study was conducted from 2016 to 2019. Please provide the exact time period (month and year).

Answer: August 2016 to July 2019.

  1. In addition to the previous comment that almost five years have passed since the results were finalized, why were the results not published earlier?

Answer: Due to lack of time and suitable team members, earlier consolidation of data and writing of manuscripts were not feasible.

  1. The authors state that the study protocol was approved by the Institutional Review Board of Chang Gung Memorial Hospital (IRB No.: 1049627A3). Please add the date of approval next to the approval number.

Answer: Thank you for your suggestions. August 1st, 2016.

  1. Please add a new paragraph in methodology (Outcomes of treatment) and describe the primary and secondary outcomes of the study.

Answer: Thank you for your comments. We have added statements in our revised manuscript.

  1. It is unclear which statistical test was used to test the normality of the data distribution. Please clarify.

Answer: Thank you for your comments. Since our sample size is nearly 200, it can be regarded as a large sample, and the distribution of the data is not skewed. Therefore, based on the central limit theorem, we present the continuous variables as Mean±SD, Parametric testing was used to compare differences between groups.

  1. The main methodological objection to this study is the lack of sample size calculation. As this is a prospective study, sample size calculation is mandatory. Please comment and clarify this point! If this was not done, the authors should clearly state why and add this to the study limitations. If a sample size calculation was performed, please include this as a separate paragraph in the methodology.

Answer: Thank you for your comments. This is a prospective study. Since there were not many patients during the admission period, there was no preset sample size at the beginning. Fortunately, we have collected more than 200 patients during the IRB period, so we can statistically analyze it.

  1. The next important objection is the presentation of the baseline characteristics of the patients. The baseline demographics of the patients and a comparison of the patients in terms of baseline demographics and other inflammatory biomarkers should be presented. It is completely inappropriate to present only the results of the biomarkers studied without providing the baseline data of the patients. Demographic data such as age, gender, comorbidities, weight, height, BMI, clinical data (pain, vomiting, body temperature, RLQ pain, rebound sensitivity) and laboratory inflammatory markers (WBC, CRP and neutrophils) should be compared between groups. In addition, these variables should also be explained in the methodology.

Answer: Thank you for your comments. We have added statements in our revised results.

  1. Tables 1 and 2 – Exact p-values (instead of p<0.05) should be reported.

Answer: Thank you for your comments. The p-values were very low, so we just presented p<0.001.

  1. The discussion is extremely extensive with many repetitions of unnecessary general data. The discussion need to be totally reorganized / revised. There is a lack of coherent comparison of the study results with the existing literature, focusing on the main objectives (especially in regards to investigated marker). Do not provide an overview of the literature in this section. Do not discuss your findings piecemeal. Focus on the results of the main objectives of the study. Write in four consecutive paragraphs (without headings) (a) summary (no data) of the findings of this study; (b) logical and coherent comparison with the existing literature focusing on the main aims; (c) limitations of the study; and (d) implications for practice/policy/research with a concluding statement.

Answer: Thank you for your suggestions. We have modified statements in our revised manuscript.

  1. The quality of the English language should be improved. The study would benefit from professional editing.

Answer: Thank you for your suggestions. The manuscript has already been sent for professional editing.  If necessary, we can send manuscript to another editor for second revision.

Comments on the Quality of English Language

The quality of the English language should be improved. The study would benefit from professional editing.

Reviewer 2 Report

Comments and Suggestions for Authors

It is a well described study for acute appendicitis and its prediction. This serum marker is interesting for this prediction. You are not referred to the cases of appendicolith, these cases were in complicated or the simple appendicitis??  Did you examine the correlation of the marker with the cases with appendicolith?

is it possible to write a short comment about the cases of appendicolith at your results?

Author Response

Reviewer 2

Comments and Suggestions for Authors

It is a well described study for acute appendicitis and its prediction. This serum marker is interesting for this prediction. You are not referred to the cases of appendicolith, these cases were in complicated or the simple appendicitis??  Did you examine the correlation of the marker with the cases with appendicolith?
is it possible to write a short comment about the cases of appendicolith at your results?

Answer: Thank you for your comments. The definite diagnosis of appendicitis was confirmed by histopathological examination of the surgical specimen. No appendicolith was noted in the histopathological in our study. Therefore, we did not examine the correlation of the marker with the cases with appendicolith.

Reviewer 3 Report

Comments and Suggestions for Authors

1.       Specify in the abstract if the study design is prospective.

2.       Specify in the abstract whether the determinations on days 1 to 3 are pre-surgical in all cases.

3.       It is a misconception to say that VCAM-1 levels are potentially useful to distinguish complicated from uncomplicated acute appendicitis based on a bivariate test. Why haven't the authors done a ROC analysis? Because they have not evaluated an AUC; an optimal cut-off point and its corresponding sensitivity and specificity?

Introduction:

1)      The introduction needs extensive revision. It is not necessary for the authors to explain how a typical appendicitis presents clinically, this is not a book chapter. Nor do the authors need to define the subtypes of appendicitis.

2)      In my opinion, a negative appendectomy rate of 3.1% is acceptable and can be framed in the context of the diagnostic error inherent to the usual clinical tools. This aspect should be qualified.

3)      The references used are poor and many of the most recent publications aimed at discriminating between complicated and uncomplicated acute appendicitis in children are not included. The authors use older nonspecific references (see ref. 57, for example) but do not review recent work evaluating cytokines in this context. for a paper that incorporates such a disproportionate number of references (64 references for an original article that is not a review is striking) this should be corrected.

Analyze and review recent systematic reviews and meta-analyses.Examples:

Hajibandeh S, Hajibandeh S, Hobbs N, Mansour M. Neutrophil-to-lymphocyte ratio predicts acute appendicitis and distinguishes between complicated and uncomplicated appendicitis: A systematic review and meta-analysis. Am J Surg. 2020 Jan;219(1):154-163. doi: 10.1016/j.amjsurg.2019.04.018. Epub 2019 Apr 27. PMID: 31056211.

Arredondo Montero J, Pérez Riveros BP, Martín-Calvo N. Diagnostic Performance of Total Platelet Count, Platelet-to-Lymphocyte Ratio, and Lymphocyte-to-Monocyte Ratio for Overall and Complicated Pediatric Acute Appendicitis: A Systematic Review and Meta-Analysis. Surg Infect (Larchmt). 2023 May;24(4):311-321. doi: 10.1089/sur.2023.013. Epub 2023 Apr 14. PMID: 37022749.

Anand S, Krishnan N, Birley JR, Tintor G, Bajpai M, Pogorelić Z. Hyponatremia-A New Diagnostic Marker for Complicated Acute Appendicitis in Children: A Systematic Review and Meta-Analysis. Children (Basel). 2022 Jul 18;9(7):1070. doi: 10.3390/children9071070. PMID: 35884054; PMCID: PMC9321702.

Arredondo Montero J, Rico Jiménez M, Martín-Calvo N. Discriminatory capacity of serum total bilirubin between complicated and uncomplicated acute appendicitis in children: a systematic review and a diagnostic test accuracy meta-analysis. Pediatr Surg Int. 2022 Dec 27;39(1):64. doi: 10.1007/s00383-022-05352-3. PMID: 36574051.

Analyze and review prospective cohorts reported recently, especially concerning serum IL-6. Examples:

Kakar M, Delorme M, Broks R, Asare L, Butnere M, Reinis A, Engelis A, Kroica J, Saxena A, Petersons A. Determining acute complicated and uncomplicated appendicitis using serum and urine biomarkers: interleukin-6 and neutrophil gelatinase-associated lipocalin. Pediatr Surg Int. 2020 May;36(5):629-636. doi: 10.1007/s00383-020-04650-y. Epub 2020 Mar 26. PMID: 32219562.

Arredondo Montero J, Antona G, Rivero Marcotegui A, Bardají Pascual C, Bronte Anaut M, Ros Briones R, Fernández-Celis A, López-Andrés N, Martín-Calvo N. Discriminatory capacity of serum interleukin-6 between complicated and uncomplicated acute appendicitis in children: a prospective validation study. World J Pediatr. 2022 Dec;18(12):810-817. doi: 10.1007/s12519-022-00598-2. Epub 2022 Sep 16. PMID: 36114365; PMCID: PMC9617836.

Zhang T, Cheng Y, Zhou Y, Zhang Z, Qi S, Pan Z. Diagnostic performance of type I hypersensitivity-specific markers combined with CRP and IL-6 in complicated acute appendicitis in pediatric patients. Int Immunopharmacol. 2023 Nov;124(Pt B):110977. doi: 10.1016/j.intimp.2023.110977. Epub 2023 Sep 27. PMID: 37774482.

4)    using nonparametric tests and using mean and standard deviation instead of median and interquartile range is a statistical inconsistency.

5)    The results section is extremely poor. Patient sociodemographics, recruitment and inclusion workflow, logistic regression analyses are not reported....

1)    The authors say "Differences between groups are presented as 95% confidence intervals (CIs)". I cannot find any data in the results expressed in this way. Gangrenous appendicitis is a complicated appendicitis. Ischemia and parietal gangrene lead almost invariably to perforation, and it is not correct to frame these patients within an isolated subgroup. The authors should speak of complicated and uncomplicated appendicitis.  In the case of table 1 they do so and it seems to me more correct.

2)    Table 1. Erratum: Uncomplicatedappendicitis.

3)      It would be interesting to establish a correlation between the serial values of this marker and other parameters such as C-reactive protein.

Comments on the Quality of English Language

Moderate editing of English language required

Author Response

Reviewer 3

Comments and Suggestions for Authors

  1. Specify in the abstract if the study design is prospective.

Answer: Thank you for your suggestions. We have added statements in our revised manuscript.

  1. Specify in the abstract whether the determinations on days 1 to 3 are pre-surgical in all cases.

Answer: Thank you for your suggestions. We have added statements in our revised manuscript.

  1. It is a misconception to say that VCAM-1 levels are potentially useful to distinguish complicated from uncomplicated acute appendicitis based on a bivariate test. Why haven't the authors done a ROC analysis? Because they have not evaluated an AUC; an optimal cut-off point and its corresponding sensitivity and specificity?

Answer: Thank you for your comments. The AUC for the VCAM-1 on day1 to day 3 may not show good enough due to the small sample sizes.

Introduction:

1) The introduction needs extensive revision. It is not necessary for the authors to explain how a typical appendicitis presents clinically, this is not a book chapter. Nor do the authors need to define the subtypes of appendicitis.

Answer: Thank you for your suggestion. We have modified statements in our revised manuscript.

2)      In my opinion, a negative appendectomy rate of 3.1% is acceptable and can be framed in the context of the diagnostic error inherent to the usual clinical tools. This aspect should be qualified.

Answer: Thank you for your suggestions. We have removed statements in our revised manuscript.

3)      The references used are poor and many of the most recent publications aimed at discriminating between complicated and uncomplicated acute appendicitis in children are not included. The authors use older nonspecific references (see ref. 57, for example) but do not review recent work evaluating cytokines in this context. for a paper that incorporates such a disproportionate number of references (64 references for an original article that is not a review is striking) this should be corrected.

Answer: Thank you for your suggestions. We have added statements in our revised manuscript.

Analyze and review recent systematic reviews and meta-analyses.Examples:

Hajibandeh S, Hajibandeh S, Hobbs N, Mansour M. Neutrophil-to-lymphocyte ratio predicts acute appendicitis and distinguishes between complicated and uncomplicated appendicitis: A systematic review and meta-analysis. Am J Surg. 2020 Jan;219(1):154-163. doi: 10.1016/j.amjsurg.2019.04.018. Epub 2019 Apr 27. PMID: 31056211.

Arredondo Montero J, Pérez Riveros BP, Martín-Calvo N. Diagnostic Performance of Total Platelet Count, Platelet-to-Lymphocyte Ratio, and Lymphocyte-to-Monocyte Ratio for Overall and Complicated Pediatric Acute Appendicitis: A Systematic Review and Meta-Analysis. Surg Infect (Larchmt). 2023 May;24(4):311-321. doi: 10.1089/sur.2023.013. Epub 2023 Apr 14. PMID: 37022749.

Anand S, Krishnan N, Birley JR, Tintor G, Bajpai M, Pogorelić Z. Hyponatremia-A New Diagnostic Marker for Complicated Acute Appendicitis in Children: A Systematic Review and Meta-Analysis. Children (Basel). 2022 Jul 18;9(7):1070. doi: 10.3390/children9071070. PMID: 35884054; PMCID: PMC9321702.

Arredondo Montero J, Rico Jiménez M, Martín-Calvo N. Discriminatory capacity of serum total bilirubin between complicated and uncomplicated acute appendicitis in children: a systematic review and a diagnostic test accuracy meta-analysis. Pediatr Surg Int. 2022 Dec 27;39(1):64. doi: 10.1007/s00383-022-05352-3. PMID: 36574051.

Analyze and review prospective cohorts reported recently, especially concerning serum IL-6. Examples:

Kakar M, Delorme M, Broks R, Asare L, Butnere M, Reinis A, Engelis A, Kroica J, Saxena A, Petersons A. Determining acute complicated and uncomplicated appendicitis using serum and urine biomarkers: interleukin-6 and neutrophil gelatinase-associated lipocalin. PediatrSurg Int. 2020 May;36(5):629-636. doi: 10.1007/s00383-020-04650-y. Epub 2020 Mar 26. PMID: 32219562.

Arredondo Montero J, Antona G, RiveroMarcotegui A, BardajíPascual C, Bronte Anaut M, Ros Briones R, Fernández-Celis A, López-Andrés N, Martín-Calvo N. Discriminatory capacity of serum interleukin-6 between complicated and uncomplicated acute appendicitis in children: a prospective validation study. World J Pediatr. 2022 Dec;18(12):810-817. doi: 10.1007/s12519-022-00598-2. Epub 2022 Sep 16. PMID: 36114365; PMCID: PMC9617836.

Zhang T, Cheng Y, Zhou Y, Zhang Z, Qi S, Pan Z. Diagnostic performance of type I hypersensitivity-specific markers combined with CRP and IL-6 in complicated acute appendicitis in pediatric patients. IntImmunopharmacol. 2023 Nov;124(Pt B):110977. doi: 10.1016/j.intimp.2023.110977. Epub 2023 Sep 27. PMID: 37774482.

4)    using nonparametric tests and using mean and standard deviation instead of median and interquartile range is a statistical inconsistency.

Answer: Since our sample size is nearly 200, it can be regarded as a large sample, and the distribution of the data is not skewed. Therefore, based on the central limit theorem, we present the continuous variables as Mean±SD, Parametric testing was used to compare differences between groups.

5)    The results section is extremely poor. Patient sociodemographics, recruitment and inclusion workflow, logistic regression analyses are not reported....

Answer: Thank you for your comments. We have added statements in our revised manuscript.

1) The authors say "Differences between groups are presented as 95% confidence intervals (CIs)". I cannot find any data in the results expressed in this way. Gangrenous appendicitis is a complicated appendicitis. Ischemia and parietal gangrene lead almost invariably to perforation, and it is not correct to frame these patients within an isolated subgroup. The authors should speak of complicated and uncomplicated appendicitis.  In the case of table 1 they do so and it seems to me more correct.

Answer: Thank you for your suggestions. We have modified statements in our revised manuscript.

2)    Table 1. Erratum: Uncomplicatedappendicitis.

Answer: Thank you for your comment. we have corrected.

3)      It would be interesting to establish a correlation between the serial values of this marker and other parameters such as C-reactive protein.

Answer: Thanks for your recommendation sincerely. We have added statements in our revised manuscript.

Reviewer 4 Report

Comments and Suggestions for Authors

Lin Wy et al. investigated whether the biomarker (VCAM-1) can be a predictor of complicated acute appendicitis. The topic is interesting, but unfortunately the study was not properly designed. I noticed several methodological objections that make this study less valuable:

-        The introduction is very long and contains several unnecessary data about acute appendicitis. It should be completely rewritten. In the introduction, the authors should address known predictors of complicated appendicitis, such as clinical scores that can differentiate between simple and complicated appendicitis. (PAS and Alvarado - https://doi.org/10.1016/j.surg.2016.06.023 or AIR score doi: 10.3390/children8040309) or laboratory biomarkers that can predict perforated from non-perforated appendicitis doi: (10.14744/tjtes.2021.83364., https://doi.org/10.3389/fphar.2022.865303 etc.)

-        The main limitation of this study is the lack of sample size calculation. Sample size calculation is mandatory for any prospective study. How did the authors know that sufficient statistical power was achieved with the sample size available?

-        The authors only presented data on the biomarkers studied. The initial parameters of the total population were to be presented and compared between the groups. In this way, the presentation of the results is very superficial

-        The discussion is below any standard. The authors should get help on how to write a discussion. The present discussion contains mainly copy/paste data from the literature. Most of the text should be removed. In the discussion, authors should compare their data with previously published data.

-        The English needs to be significantly improved

Comments on the Quality of English Language

 The English needs to be significantly improved

Author Response

Reviewer 4

Comments and Suggestions for Authors

Lin Wy et al. investigated whether the biomarker (VCAM-1) can be a predictor of complicated acute appendicitis. The topic is interesting, but unfortunately the study was not properly designed. I noticed several methodological objections that make this study less valuable:

-        The introduction is very long and contains several unnecessary data about acute appendicitis. It should be completely rewritten. In the introduction, the authors should address known predictors of complicated appendicitis, such as clinical scores that can differentiate between simple and complicated appendicitis. (PAS and Alvarado - https://doi.org/10.1016/j.surg.2016.06.023 or AIR score doi: 10.3390/children8040309) or laboratory biomarkers that can predict perforated from non-perforated appendicitis doi: (10.14744/tjtes.2021.83364., https://doi.org/10.3389/fphar.2022.865303 etc.)

Answer: Thank you for your comment. We have revised our introduction based on your suggestion and added the two papers as our reference.

-        The main limitation of this study is the lack of sample size calculation. Sample size calculation is mandatory for any prospective study. How did the authors know that sufficient statistical power was achieved with the sample size available?

Answer: This is a prospective study. Since there were not many patients during the admission period, there was no preset sample size at the beginning. Fortunately, we have collected more than 200 patients during the IRB period, so we can statistically analyze it.

-        The authors only presented data on the biomarkers studied. The initial parameters of the total population were to be presented and compared between the groups. In this way, the presentation of the results is very superficial

Answer: This is the first study to identify the biomarker VCAM-1 in predicting complicated appendicitis in children, so the total population were to be presented and compared between the groups. In the future, our research will be more rigorous.

-        The discussion is below any standard. The authors should get help on how to write a discussion. The present discussion contains mainly copy/paste data from the literature. Most of the text should be removed. In the discussion, authors should compare their data with previously published data.

Answer: Thank you for your comments. We have revised our manuscript based on your suggestions.

-        The English needs to be significantly improved

    Answer: Thank you for your comment. We have sent our paper for English editing by native english speakers.

Round 2

Reviewer 1 Report

Comments and Suggestions for Authors

The authors have made some revisions, but the improvement in scientific value is not satisfactory for the following reasons:

- The statistical analysis is inadequate. The data should be tested for normal distribution.

- The authors did not calculate the sample size, which is one of the major shortcomings of this study and may be an important source of bias. In any prospective study, sample size calculation is mandatory.

-The presentation of the baseline characteristics of the patients is still inadequate. It is completely inappropriate to present only the results of the biomarkers studied without providing the baseline data of the patients. Demographic data such as age, gender, comorbidities, weight, height, BMI, clinical data (pain, vomiting, body temperature, RLQ pain, rebound sensitivity) and inflammatory markers from the laboratory (WBC, CRP and neutrophils) should be compared between the groups. In addition, these variables should also be explained in the methodology. The authors should create a table with all the demographics listed, they only listed inflammatory markers.

- The discussion is still poor and contains a lot of unnecessary general details. The discussion is extremely lengthy and contains a lot of repetition of unnecessary data. All general statements about appendicitis should be deleted and the discussion should focus on inflammatory biomarkers in acute appendicitis.

- Several biomarkers were suggested, but the authors did not even mention them (E.g. LRG-1, NGAL, PTX-3, etc.).

Comments on the Quality of English Language

Some improvements have been made to the English, but a moderate revision of the English language is still required.

Author Response

(Reviewer 1

Comments and Suggestions for Authors

The authors have made some revisions, but the improvement in scientific value is not satisfactory for the following reasons:

- The statistical analysis is inadequate. The data should be tested for normal distribution.

Answer: Thank you for your suggestions. Data that do not follow a normal distribution, so we present the continuous variables as Median, Nonparametric statistics, Mann-Whitney U Test was used to compare differences between groups.

- The authors did not calculate the sample size, which is one of the major shortcomings of this study and may be an important source of bias. In any prospective study, sample size calculation is mandatory.

Answer: Thank you for your suggestions. Since our sample size is nearly 200, it can be regarded as a large sample, and the comparison of levels in VCAM-1 between the two groups showed highly significant differences. Although we did not calculate the sample size, we think our sample size nearly 200 could be sufficient to achieve statistical significance for this survey.

-The presentation of the baseline characteristics of the patients is still inadequate. It is completely inappropriate to present only the results of the biomarkers studied without providing the baseline data of the patients. Demographic data such as age, gender, comorbidities, weight, height, BMI, clinical data (pain, vomiting, body temperature, RLQ pain, rebound sensitivity) and inflammatory markers from the laboratory (WBC, CRP and neutrophils) should be compared between the groups. In addition, these variables should also be explained in the methodology. The authors should create a table with all the demographics listed, they only listed inflammatory markers.

Answer: Thank you for your suggestions. We have added statements as Table 1 in our revised manuscript.

- The discussion is still poor and contains a lot of unnecessary general details. The discussion is extremely lengthy and contains a lot of repetition of unnecessary data. All general statements about appendicitis should be deleted and the discussion should focus on inflammatory biomarkers in acute appendicitis.

Answer: Thank you for your suggestions. We have revised our discussion section in our revised manuscript.

- Several biomarkers were suggested, but the authors did not even mention them (E.g. LRG-1, NGAL, PTX-3, etc.).

Answer: Thank you for your suggestions. We have added the related statements about LRG-1, NGAL, PTX-3 in our revised manuscript.

Reviewer 3 Report

Comments and Suggestions for Authors

The manuscript has improved ostensibly at the expense of the changes introduced. However, there is still a major issue that needs to be solved:

I find it inconsistent for the authors to say that they have a sample of 200 patients and therefore assume a normal distribution without performing specific tests (K-G, S-W) and, on the other hand, to say that since they expect poor results in the ROC analysis, they do not include them. This is called publication bias. 

1.1. Please evaluate specifically and by statistical tests the normal or non-normal distribution of variables with a continuous quantitative distribution and evaluate Levene's homogeneity of variances. If you want to express the results in mean + sd, please clarify that it is for the reader's convenience. Do not omit this analysis. The normal or non-normal distribution does not depend only on the sample size, but also on the biological behavior of the variable under study. 

1.2. It is absolutely pertinent to perform a logistic regression analysis (ROC) at least for the initial measurement and comparison of the two main groups. It is important to report the AUC values with their 95% CI so that readers can understand the true diagnostic and discriminative performance of this molecule. I recommend including the graphical plots (lroc) of the analyses and commenting in some detail on this aspect. The presence of statistically significant differences in the serum values of this biomarker is by no means synonymous with adequate diagnostic/discriminative performance.

For the rest, and once this aspect is solved, the work deserves acceptance on my opinion. 

Comments on the Quality of English Language

Minor editing. 

Author Response

Reviewer 3

Comments and Suggestions for Authors

The manuscript has improved ostensibly at the expense of the changes introduced. However, there is still a major issue that needs to be solved:

I find it inconsistent for the authors to say that they have a sample of 200 patients and therefore assume a normal distribution without performing specific tests (K-G, S-W) and, on the other hand, to say that since they expect poor results in the ROC analysis, they do not include them. This is called publication bias. 

1.1. Please evaluate specifically and by statistical tests the normal or non-normal distribution of variables with a continuous quantitative distribution and evaluate Levene's homogeneity of variances. If you want to express the results in mean + sd, please clarify that it is for the reader's convenience. Do not omit this analysis. The normal or non-normal distribution does not depend only on the sample size, but also on the biological behavior of the variable under study. 

Answer: Thank you for your suggestions. We have revised the results to express the data as median (25-75 %) in our revised manuscript.

1.2. It is absolutely pertinent to perform a logistic regression analysis (ROC) at least for the initial measurement and comparison of the two main groups. It is important to report the AUC values with their 95% CI so that readers can understand the true diagnostic and discriminative performance of this molecule. I recommend including the graphical plots (lroc) of the analyses and commenting in some detail on this aspect. The presence of statistically significant differences in the serum values of this biomarker is by no means synonymous with adequate diagnostic/discriminative performance.

Answer: Thank you for your suggestions. We have added ROC analysis in our revised manuscript.

Reviewer 4 Report

Comments and Suggestions for Authors

The authors failed to address my comments. Baseline data (clinical/demographic) were not presented or compared between groups. Since this is a prospective study, the sample size calculation is missing. How do the authors know that this sample is sufficient to achieve statistical significance? For such a common pathology, this sample is weak. The discussion is poor and I see no improvement. English needs to be significantly improved.

Comments on the Quality of English Language

English needs to be significantly improved

Author Response

Reviewer 4

Comments and Suggestions for Authors

The authors failed to address my comments. Baseline data (clinical/demographic) were not presented or compared between groups. Since this is a prospective study, the sample size calculation is missing. How do the authors know that this sample is sufficient to achieve statistical significance? For such a common pathology, this sample is weak. The discussion is poor and I see no improvement. English needs to be significantly improved.

Answer: Thank you for your suggestions. Since our sample size is nearly 200, it can be regarded as a large sample, and the comparison of levels in VCAM-1 between the two groups showed highly significant differences. Although we did not calculate the sample size, we think our sample size nearly 200 could be sufficient to achieve statistical significance for this survey.